# Prefrontal cortical activity predicts the occurrence of nonlocal hippocampal representations during spatial navigation

**Jai Y. Yu**[1]*, **Loren M. Frank**[2]

**1** Department of Psychology, Institute for Mind and Biology, Neuroscience Institute, University of Chicago, Chicago, Illinois, United States of America, **2** Howard Hughes Medical Institute, Kavli Institute for Fundamental Neuroscience, Departments of Physiology and Psychiatry, University of California, San Francisco, San Francisco, California, United States of America

* jaiyu@uchicago.edu

**Data Availability Statement:** Data used for this manuscript can be accessed at https://crcns.org/data-sets/hc/hc-13/about-hc-13 and in the Supporting Information.

## Abstract

The receptive field of a neuron describes the regions of a stimulus space where the neuron is consistently active. Sparse spiking outside of the receptive field is often considered to be noise, rather than a reflection of information processing. Whether this characterization is accurate remains unclear. We therefore contrasted the sparse, temporally isolated spiking of hippocampal CA1 place cells to the consistent, temporally adjacent spiking seen within their spatial receptive fields ("place fields"). We found that isolated spikes, which occur during locomotion, are strongly phase coupled to hippocampal theta oscillations and transiently express coherent nonlocal spatial representations. Further, prefrontal cortical activity is coordinated with and can predict the occurrence of future isolated spiking events. Rather than local noise within the hippocampus, sparse, isolated place cell spiking reflects a coordinated cortical–hippocampal process consistent with the generation of nonlocal scenario representations during active navigation.

## Introduction

The concept of a receptive field [1–3] provides a fundamental model for how neural spiking can convey information about features in the external environment. In the hippocampus, many cells show spatially tuned receptive fields [4,5]. The spiking rate of these "place cells" rises and then falls as an animal traverses specific locations in an environment. In linear environments, the animal's movement direction can also modulate spiking [6–10], resulting in location and direction-specific activity. Locations and directions with high spiking rates are defined as a cell's "place fields" [5,6], and place field–associated spiking of place cells conveys sufficient spatial information to estimate the animal's location with high accuracy [11–14].

Although the majority of place cell spiking occurs when an animal is moving within the cell's place field(s), occasional spiking occurs when the animal is at locations outside the field(s) [5,6,15–17]. These "isolated" spiking events can occur during movement and are distinct from sparse spiking observed during sharp-wave ripples (SWRs) seen during immobility [18]. Importantly, isolated spikes are not locked to specific locations. As a result, standard analyses

**Funding:** This work was supported by a Jane Coffin Childs Memorial Fund for Biomedical Research postdoctoral fellowship (J.Y.Y.), the Howard Hughes Medical Institute, the Kavli Institute for Fundamental Neuroscience, and University of California Office of the President Lab Fees Award #LF-12-237680 (L.M.F.). The funders had no role in study design, data collection and analysis, decision to publish, or preparation of the manuscript.

**Competing interests:** The authors have declared that no competing interests exist.

**Abbreviations:** AP, Anterior-Posterior; DV, Dorsal-Ventral; GLM, generalized linear model; LFP, local field potential; MAE, mean absolute error; ML, Medial-Lateral; PBS, phosphate buffered saline; PFC, prefrontal cortex; SWR, sharp-wave ripple.

that average activity across many passes through the same location [13,15,19–22] effectively exclude these spikes from further consideration. Whether these spikes reflect unreliable, noisy processes that merit exclusion or whether they instead reflect coherent, meaningful signals remains unknown.

Noise in neural networks can arise from stochastic cellular events that cause the membrane voltage to occasionally exceed the action potential threshold, even without upstream input [23,24]. While the spatially and directionally selective inputs to a place cell raise the membrane voltage closer to the action potential threshold when an animal approaches the cell's place field [25,26], stochastic events causing occasional increases in membrane potential could result in spiking outside of a cell's place field. However, previous observations indicate that at least some spiking outside of a cell's typical place fields reflect mnemonic processes rather than noise. CA1 and CA3 place cells can emit spikes outside of their place fields as an animal approaches choice points [27,28] and during vicarious trial and error [27] or when an animal is traveling in the opposite direction over a location with a place field [28]. These events are hypothesized to reflect noncurrent scenarios, such as simulating possible future scenarios when a decision needs to be made [28,29].

How can we determine whether isolated spiking outside of a place cell's spatially and directionally tuned receptive field reflects information processing in the hippocampal circuit as opposed to activity that does not reflect information processing or noise? Spiking due to stochastic cellular events is expected to be local to individual neurons. By contrast, spiking associated with information processing would be expected to covary in a consistent manner across neurons in both local and distributed networks [30]. Thus, if spiking outside of the classical place field conveys information, we would expect it to (1) be coordinated across multiple hippocampal neurons; (2) contain coherent spatial information; and (3) be coordinated with activity outside the hippocampus.

We therefore examined spiking both within the hippocampus and across the hippocampus and prefrontal cortex (PFC), focusing on activity during movement. PFC is anatomically connected to the hippocampus through both direct and indirect projections [31–33], and coordinated activity across these networks reflects their engagement during memory processing [34–36]. For example, network level coherence between PFC and hippocampus increases during periods when memory retrieval occurs [37–44]. Whether PFC activity differs systematically at the time of isolated spiking in the hippocampus remains unknown.

Our examination of isolated spiking of place cells revealed that these events reflect the coherent activation of hippocampal representations of physically distant locations and that these events are coordinated with ongoing activity in the PFC. These findings suggest that isolated spikes are a signature of distributed and coherent information processing in the brain.

## Results

In order to understand the extent of isolated spiking during active behavior and to identify a potential function of this activity, we took an unbiased approach where we surveyed CA1 place cell spiking across all movement periods (animal speed >2 cm/s) as animals performed a spatial navigation task in a complex environment with multiple linear track segments [45,46] (Fig 1A and 1B). In the hippocampus, the temporal structure of spiking during locomotion is strongly influenced by the endogenous approximately 8-Hz theta rhythm [47], and bouts of higher rate spiking corresponding to place field traversals spanned multiple, adjacent cycles of theta (Fig 1C). As expected, we also observed isolated spikes where a neuron would be silent for many theta cycles, emit a small number of spikes on a single theta cycle, and then return to being silent (Fig 1D) [15,19,20,27].

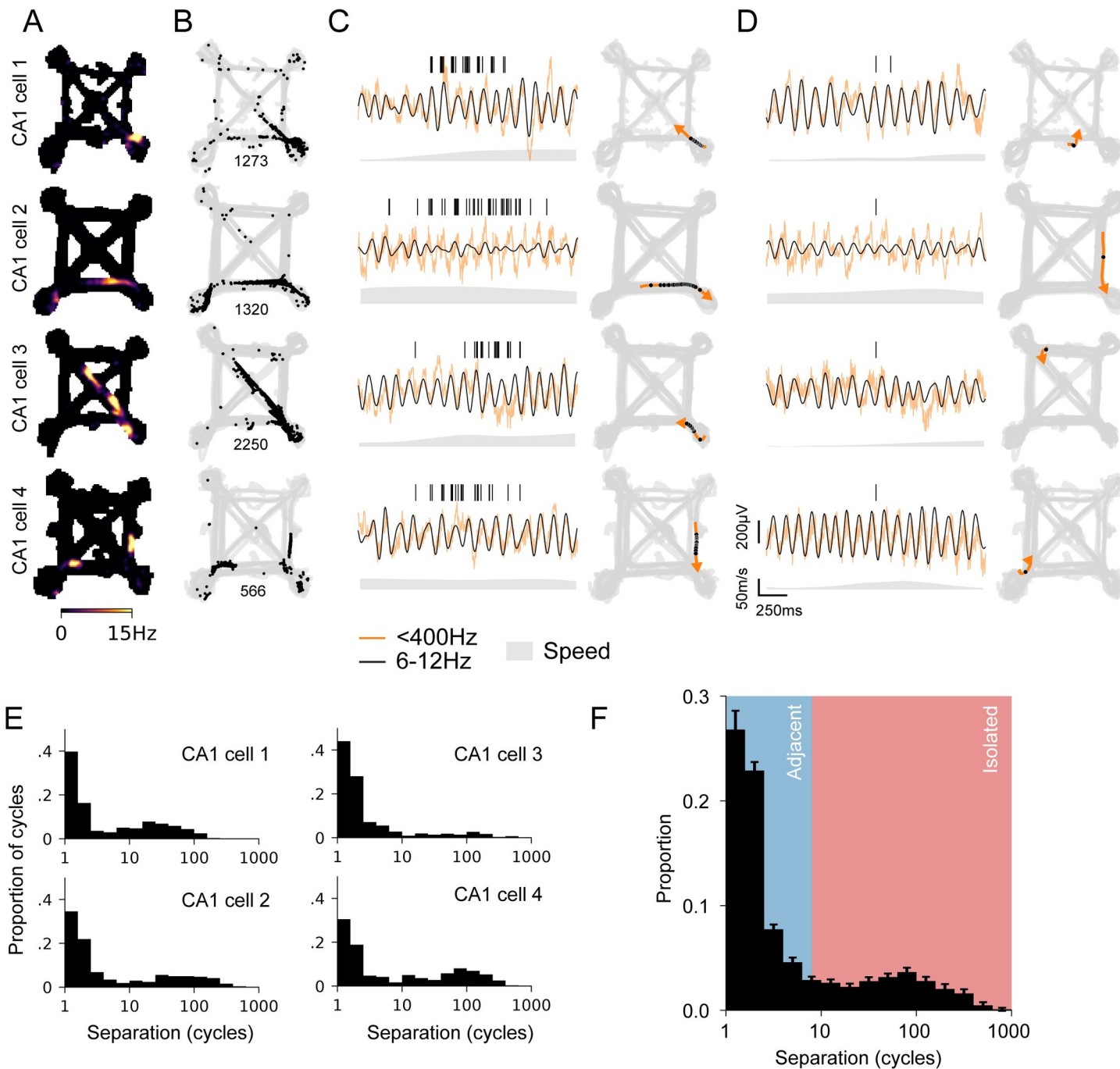

**Fig 1. Isolated and adjacent spiking activity of hippocampal CA1 place cells. (A)** Occupancy normalized spiking rate maps for spiking activity during active movement (animal speed >2 cm/s) across behavior sessions for each day for 4 example CA1 cells. **(B)** Location of spiking (black dots) and animal trajectory (gray) for rate maps in A. Spike count shown below each panel. **(C)** Spike raster and corresponding location for a bout of spiking activity over adjacent theta cycles. Raw (orange) and theta frequency filtered (black) LFP are shown below the spike raster. The corresponding location of the animal's trajectory (orange line and arrow) and the spikes (black dots) on the maze for the bout are shown on the right. **(D)** Spike raster and corresponding location for spiking isolated from other spiking activity. The corresponding location on the maze for the bout is shown on the right. **(E)** Distribution of mean separation between theta cycles with spiking. Separation is defined as the mean cycle count to the 3 nearest neighbor cycles with spiking. **(F)** Population distribution of mean separation between theta cycles with spiking (*n* = 298 cells). LFP, local field potential.

The standard approach to defining place field spiking relies on averaging spiking rates across many traversals of a location. This average provides a useful experimental summary of spiking, but information averaged across traversals is not directly available to downstream neurons. We therefore we used a criterion to distinguish between "adjacent" and "isolated" spiking based on the local temporal organization of spiking. Specifically, given the importance of theta in organizing hippocampal activity [47–49], we calculated the interval between neighboring theta cycles with spiking (in cycles, mean of nearest 3) (Fig 1E, S1A Fig). As expected, most spike-containing theta cycles are near another spike-containing cycle. The remaining spike-containing theta cycles are separated from neighboring spike-containing cycles by up to hundreds of cycles, reflecting their temporal isolation. When plotted on a log scale, the underlying distribution was bimodal, and based on this distribution, we chose a threshold of 8 cycles of mean separation to each theta cycle with spiking to define "adjacent" or "isolated" activity ($n$ = 298 cells, Fig 1F). This method identifies spiking on individual theta cycles and also spiking on a small number of nearby cycles that are nevertheless isolated from periods of adjacent spiking.

This separation captured intuitive notions of within- and extra-place field activity: Adjacent activity was spatially concentrated and had high spiking rates, as expected from place-field spiking (Fig 2A). By contrast, isolated activity was spatially sparse and lacked the high spiking rates observed for place field activity. Nonetheless, isolated spiking represented 17 ± 1.8% (median ± 95% CI) of spikes included for analysis (S1B Fig). As expected, while individual adjacent spikes tended to occur at locations close to other adjacent spikes (median$_{adjacent-adjacent}$ = 0.49 cm; Fig 2B), isolated spikes tended to occur at more distant locations (median$_{isolated-adjacent}$ = 8.98 cm, $p = 6.02 \times 10^{-76}$; Fig 2B). In cases where isolated spikes occurred at locations close to adjacent spiking, these isolated spikes typically occurred on a different trial (Fig 2C, median$_{adjacent-adjacent}$ = 0.18 seconds, median$_{isolated-adjacent}$ = 28.23 seconds, $p = 3.10 \times 10^{-57}$; trials are approximately 7.5 seconds long [45]). Consistent with previous findings, these isolated spikes most often occurred when the animal was traveling in the opposite direction compared with adjacent spiking (Fig 2D, median$_{adjacent-adjacent}$ = 0.22˚, median$_{isolated-adjacent}$ = 158.9˚, $p = 4.66 \times 10^{-76}$) [6–9,28]. Importantly, isolated spiking was not well explained by a cell's overall tendency to show more spatially diffuse representations since the spatial distribution properties of adjacent spiking are not correlated with the proportion of isolated spiking (S2 Fig). We also verified that isolated spikes, although sparsely emitted, were very unlikely to be spike cluster assignment errors (S3 Fig).

As expected, isolated spiking was also highly concentrated within the later phases of each theta cycle (Fig 3A and 3B). Place field–associated spiking displays strong phase coupling to the hippocampal theta rhythm, where the maximum probability of spiking occurs in earlier phases near the trough of theta [47,50]. Later phases correspond to times where inhibition is lower, and, thus, activity outside the main place field could be generated [26,49]. Isolated spiking was also more tightly phase locked to theta compared with adjacent spiking (Fig 3C). This was true both for isolated spikes that occurred close to locations where adjacent spiking was seen and for isolated spikes that occurred far from those locations (S4 Fig), indicating that isolated spiking has similar network coupling properties irrespective of their spatial proximity to the cell's adjacent spiking.

We also ensured isolated spiking was not associated with SWRs, which are transient network oscillations observed in the local field potential (LFP) and are predominantly found when the animal is moving slowly or is immobile [18]. This was done by excluding SWRs from our analyses (see Methods) and independently confirming the isolated spiking events did not have the spectral signature of SWRs. The LFP associated with excluded spiking showed a network spectral signature consistent with SWRs (S5A Fig, left column; S5B Fig), with power

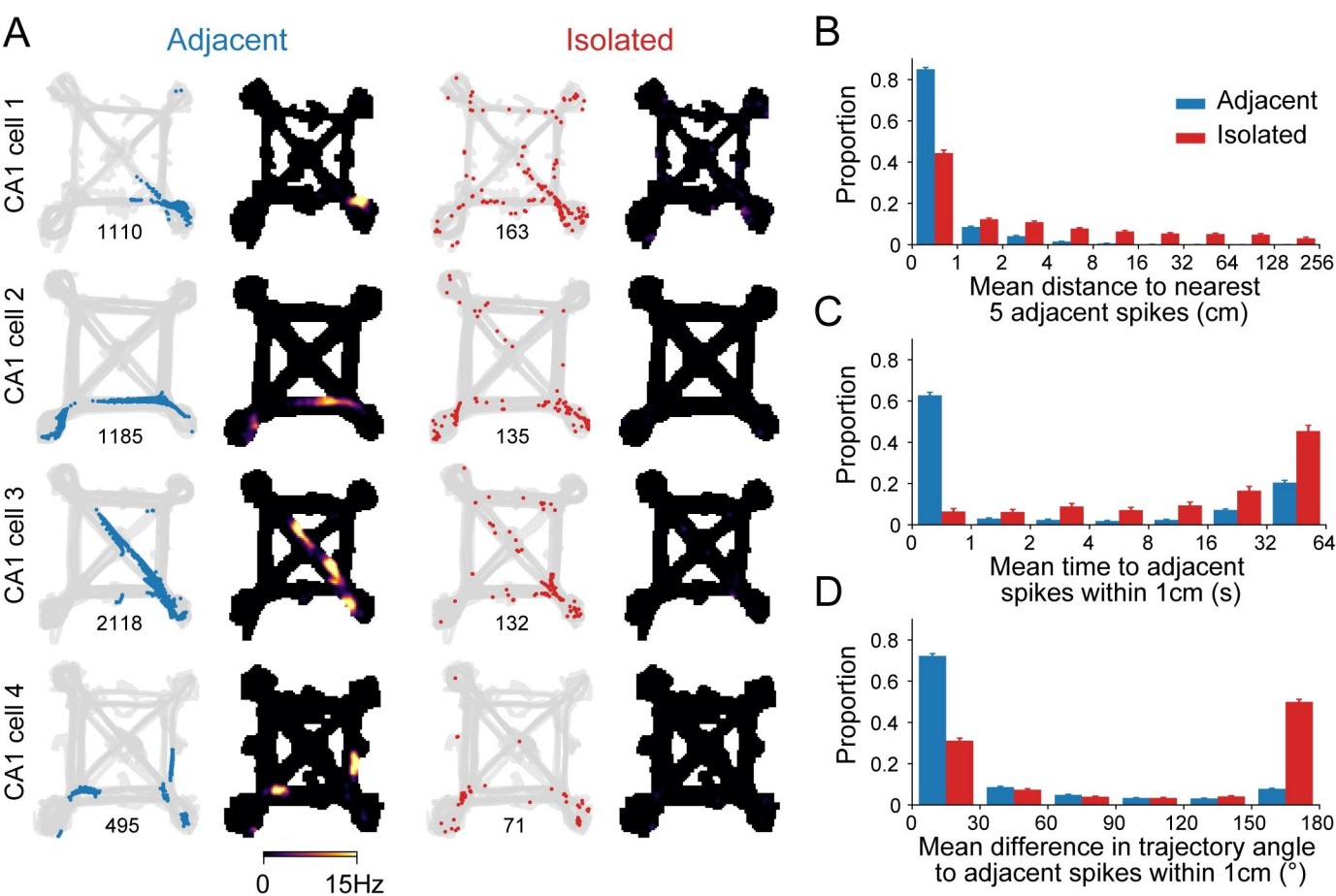

**Fig 2. Spatial and temporal separation between isolated and adjacent spiking. (A)** Location of spiking classified as adjacent or isolated activity for the 4 example cells in Fig 1A. Spike count shown below each panel. **(B)** Mean distance on the maze from one spike (adjacent: blue or isolated: red) to its nearest 5 neighboring adjacent spikes. The distance is the shortest path on the maze between 2 spikes. Wilcoxon rank sum test: $p = 6.02 \times 10^{-76}$. **(C)** Mean separation in time between one spike and other adjacent spikes that occur at locations on the maze within 1 cm. Wilcoxon rank sum test: $p = 3.10 \times 10^{-57}$. **(D)** Mean difference in the trajectory vector between one spike and other adjacent spikes within 1 cm. Wilcoxon rank sum test: $p = 4.66 \times 10^{-76}$. (B–D) Histogram shows mean ± SEM for across 247 cells.

in the slow gamma (approximately 30 Hz) and ripple frequencies (approximately 150 to 250 Hz). In contrast, the LFP associated with isolated spiking shows a different network spectral signature, with power in the theta band [50] (S5A Fig, right column; S5B Fig). Indeed, the network spectral signature of isolated spiking is very similar to the LFP associated with adjacent spiking and even has slightly higher theta power (S5A Fig, center column; S5B Fig).

Recent findings from our group indicated that spiking related to possible future locations or opposite directions of travel can occur in animals traveling at high speeds and in the absence of overt deliberative behaviors [28]. We replicated these findings for isolated spikes: Isolated activity was not more frequent around choice point locations (Fig 4A), nor were there differences in the speed (Fig 4B) or angular acceleration (Fig 4C) of the animal at times of isolated as compared to adjacent spiking. Thus, isolated spiking is not restricted to specific active behavioral states or locations, such as path choice points. We next examined the relationship between isolated spiking and task behavior. We reasoned that if these events reflect task-related cognitive processes, we may see differences in the rate of isolated spiking across different phases of a behavior session. For the early trials in a behavior session, when the animal explored the environment in order to find the current reward locations, its performance is

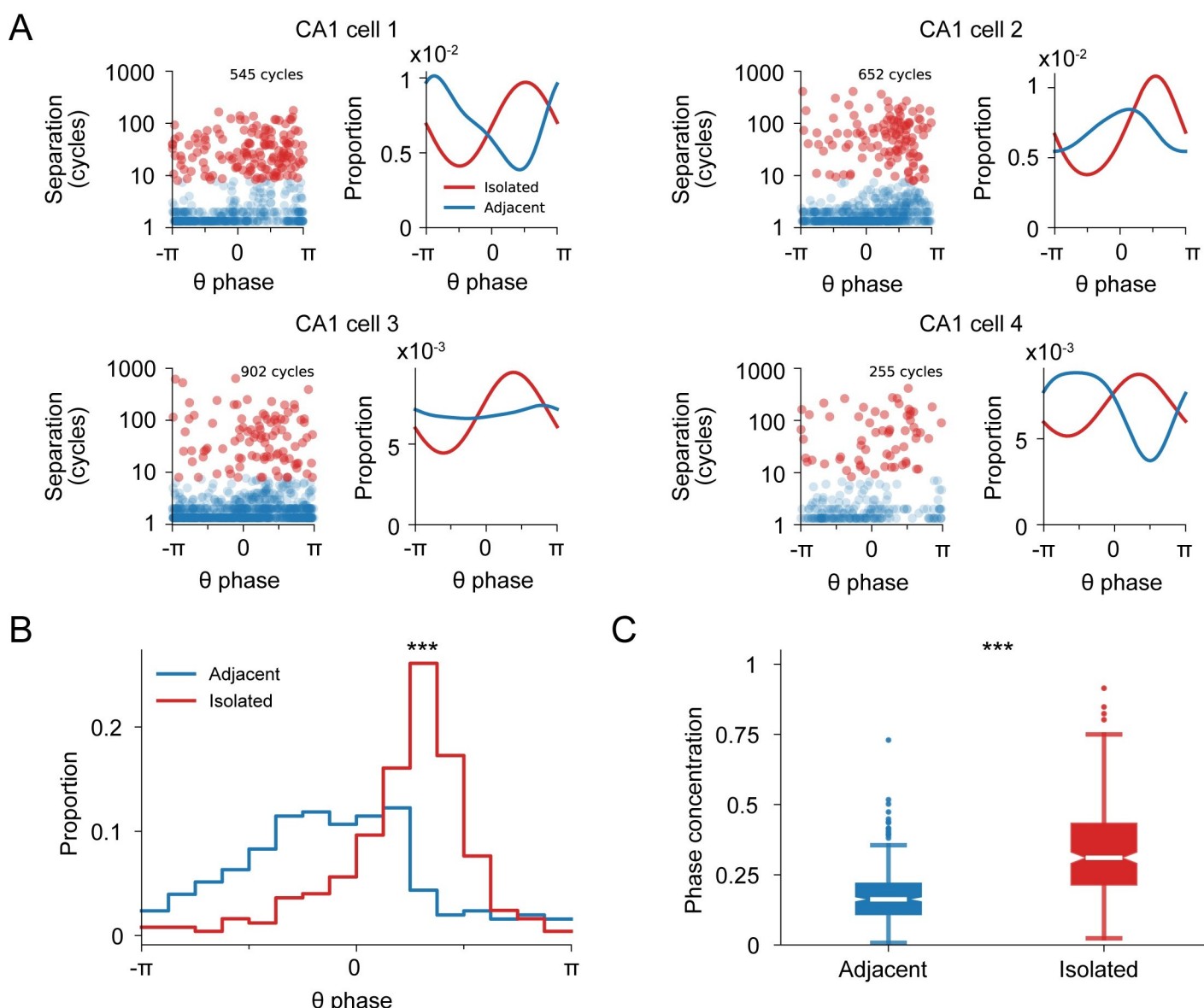

**Fig 3. Isolated and adjacent spiking activity show distinct phase locking to hippocampal theta oscillations.** (**A**) Theta cycle separation versus mean spike theta phase preference. A separation threshold of 8 cycles between isolated and adjacent classification is based on Fig 1. Histogram shows the mean spiking phase for each theta cycle. Examples correspond to the 4 cells from Fig 1. Circular median test between the isolated and adjacent distributions: top left: $p = 1.4 \times 10^{-5}$; top right: $p = 8.9 \times 10^{-8}$; bottom: left $p = 2.4 \times 10^{-4}$; bottom right: $p = 5.3 \times 10^{-2}$. (**B**) Mean theta phase preference distribution for adjacent and isolated spiking for the CA1 cell population ($n = 247$ cells). Circular median test: $p = 0$. (**C**) Mean theta phase concentration distribution for adjacent and isolated spiking for the CA1 cell population ($n = 247$ cells). Wilcoxon rank sum test: $p = 2.09 \times 10^{-28}$.

low. In the last trials of a session, when the animal has correctly identified the reward locations, its performance becomes high (Fig 4D). The difference in performance is not a reflection of unfamiliarity with the task since the median trial durations are comparable (Fig 4E). During the first 5 trials, we found a significantly higher rate of isolated spiking compared with the last 5 trials of a behavior session (Fig 4F). This was not the case for adjacent spiking (Fig 4G), and overall place representations are stable across each behavior session. Thus, these observations suggest that isolated spiking is associated with discovering new reward rules during early trials of a new behavior session.

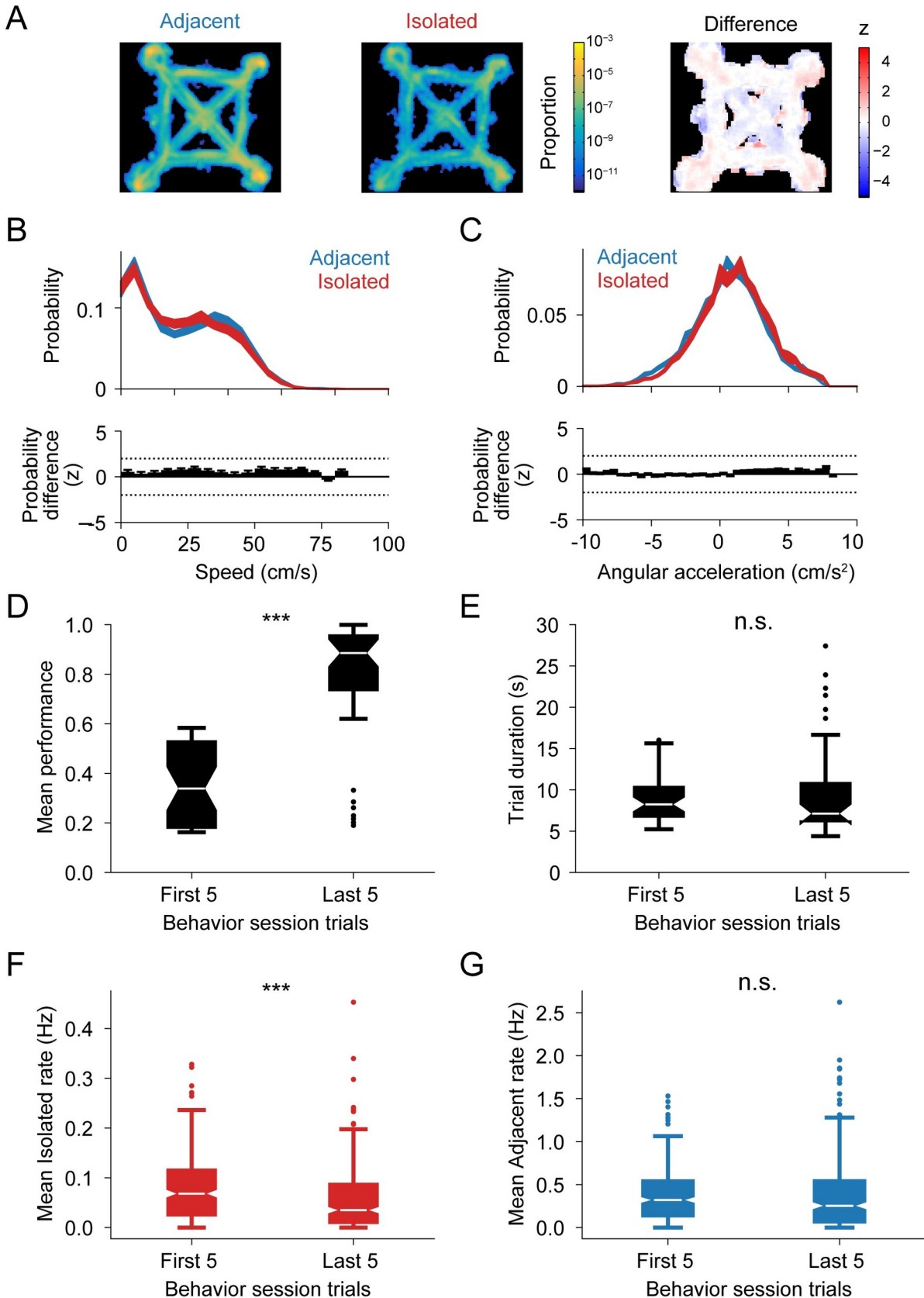

**Fig 4. Spatial distribution and behavioral correlates of isolated and adjacent spiking. (A)** Normalized spatial distribution of theta cycles with adjacent (left) or isolated (center) spiking. Normalized difference between the spatial distributions (right). **(B)** Distribution of animal speed (mean ± SEM) at the time of adjacent or isolated activity (top). Significance of the difference (z) between the 2 distributions as determined using a permutation test (bottom). Dotted lines indicate ± 2 z. **(C)** Distribution of animal angular acceleration (mean ± SEM) at the time of adjacent or isolated activity (top). Significance of the difference (z) between the 2 distributions as determined using a permutation test (bottom). Dotted lines indicate ± 2 z. **(D)** Mean performance for the first and last 5 trials of each behavior session ($n$ = 38). Wilcoxon signed rank test: $p = 7.74 \times 10^{-8}$. **(E)** Median duration for the first and last 5 trials of each behavior session ($n$ = 38). Wilcoxon signed rank test: $p = 0.89$. **(F)** Mean isolated spiking rate for the first and last 5 trials of each behavior session ($n$ = 247 cells). Wilcoxon signed rank test: $p = 9.79 \times 10^{-9}$. **(G)** Mean adjacent spiking rate for the first and last 5 trials of each behavior session ($n$ = 247 cells). Wilcoxon signed rank test: $p = 0.91$.

Individual hippocampal place cells can be active in cell assemblies [51] that show temporally correlated activity on multiple timescales [52–54]. High spiking rate activity of these cell assemblies, typically associated with place fields, express information about current location of the animal. We therefore asked whether isolated spiking reflects coordinated activity between CA1 cells and sought to identify their corresponding spatial representations. Specifically, we knew that spiking during the late phases of theta is associated with the expression of nonlocal representations, including to be visited locations or locations previously visited [28,55,56]. Given that isolated spikes occur at late phases of theta and are separated from adjacent spiking both in time and space, could isolated spiking reflect the transient activity of cell assemblies with place field activity in another part of the environment? If so, then we would expect that pairs of neurons that are coactive during periods of adjacent spiking, corresponding to cells that are likely to have overlapping place fields, would also be coactive within a theta cycle containing isolated spiking events. We examined this possibility by using an approach that has been used to demonstrate reactivation of nonlocal spatial representations during SWRs, where a pair of place cells is more likely to spike together if their place fields overlap [57,58] (Fig 5A). First, we calculate the likelihood of co-spiking for a pair of place cells that had isolated spiking within the same theta cycle. We then quantified the overlap in their adjacent spiking activity. We found that cells that fired together during periods of adjacent spiking were also more likely to fire together during isolated spiking events. Across the population, lower lags in spiking during adjacent activity were correlated with greater co-spiking during isolated events (Fig 5B, R = −0.28, $R^2$ = 0.077, $p = 6.40 \times 10^{-9}$). These findings support the notion that isolated spikes may reflect the spiking of cell assemblies with spatial representations for locations away from the animal or in the direction of travel opposite to the current direction.

We next asked whether isolated spiking events reflect spontaneous activation of cell assemblies in CA1 or coordinated activity with other functionally connected networks. We examined simultaneously recorded activity in PFC, a region that is connected with CA1 by mono- and multisynaptic pathways [31–33]. Given the anatomical connectivity between these regions, evidence of spiking coordination between hippocampus and PFC would strongly suggest that these events are not the result of spontaneous activation of local cell assemblies in the hippocampus but instead reflect coherent and structured activity across brain regions. An example of such hippocampal–cortical engagement occurs during SWRs, where hippocampal reactivation is accompanied by the coordinated reactivation of cortical representations [45,46,59–62]. If such coordination is seen around the times of isolated spikes, we should be able to identify PFC neurons that spike differently around times of isolated activity in the hippocampus than at comparable periods during which isolated spiking was not observed.

Although isolated spiking does not occur at specific locations, we can use the times of isolated spiking events as reference points to look for coordination with PFC activity. We first selected theta cycles with isolated spiking for a given CA1 cell. Next, we found matching theta cycles from other times when the animal was moving through the same locations in the same

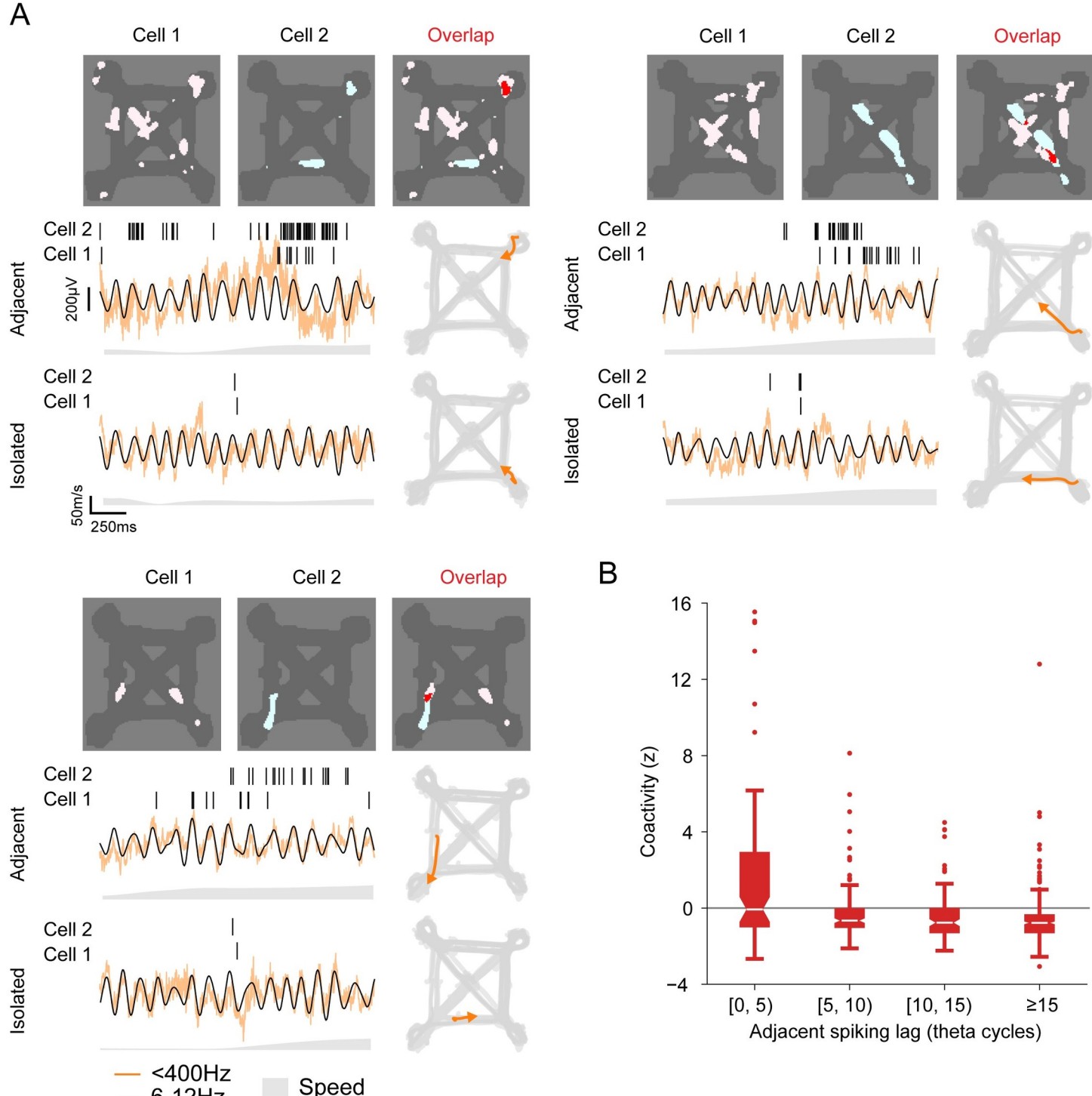

**Fig 5. Reactivation of spatiotemporal place field activity relationships during theta cycles with isolated spiking activity.** (A) Three pairs of CA1 cells with overlapping adjacent activity. The place fields (occupancy normalize spiking rate >5 Hz) for each cell as well as their spatial overlap are shown. Example spiking bouts of adjacent and isolated activity are shown with raw and theta frequency band filtered LFP. The animal's trajectory (orange line and arrow) on the maze for each bout is shown on the right. (B) Normalized coactivity (z) for CA1 cell pairs during theta cycles with isolated activity (*n* = 425 pairs) grouped by the mean separation in time (mean lag) between their adjacent activity (R = −0.28, R² = 0.077, *p* = 6.40 × 10⁻⁹). LFP, local field potential.

direction at a similar speed, but where the CA1 cell was not active (e.g., did not have isolated spiking) (S6 Fig). This was possible because, in our task, the animal traversed a given location multiple times, providing a pool of theta cycles, of which only a subset contained isolated spiking. Importantly, none of the matching cycles contained adjacent spiking, confirming that the isolated spiking events were not simply events on the edge of a place field. We then compared the spiking of simultaneously recorded PFC neurons between cycles with isolated activity and these matched control cycles (S6A Fig). We note that theta coordinates activity in hippocampal–cortical networks [40, 41], allowing us to continue to use theta cycles as the temporal reference to relate activity across structures.

We found multiple PFC cells whose spiking rate differed depending on whether there was an associated period of isolated spiking for a given CA1 cell (Fig 6B). We expect that only a small fraction of PFC cells would show a significant difference in spiking relative to the isolated spiking of a given CA1 cell, but, nonetheless, across the population ($n$ = 2,798 PFC–CA1 cell pairs), the difference in PFC firing rates between isolated and matched control periods was significantly larger than the permutation control (Fig 6C, S7 Fig). This difference indicates coordination between CA1 and PFC around the time of CA1 isolated activity. Interestingly, this coordination was not limited to the specific isolated theta cycle: The difference remained significant even in a window of 8 to 12 theta cycles before the isolated spiking event, indicating that PFC activity could play a causal role in driving isolated spiking events in the hippocampus. In addition, we found significant PFC spiking rate differences remained after the occurrence of CA1 isolated activity (Fig 6D). We also verified that this coordination could not be explained by the higher rate of isolated spiking found early in each epoch: Overall, there was no significant difference in the time interval between pairs of isolated cycles and pairs of matched cycles (S6E Fig). Thus, any coordination found cannot be explained by isolated cycles being closer together in time than their matched counterparts.

If these differences signify coordinated activity, the ensemble activity of PFC neurons should predict the future occurrence of hippocampal isolated activity (Fig 7A). To test that prediction for a given CA1 cell, we used the spiking activity from simultaneously recorded PFC ensembles (median $n$ = 20, IQR = 8 PFC cells per CA1 cell) to build cross-validated generalized linear models (GLMs) with elastic net regularization. We compared the ability of the models to predict the occurrence of isolated activity relative to a permutation control (see Methods; S8 Fig). We then carried out that analysis for CA1 cells ($n$ = 158) with isolated spiking.

We found that PFC activity can predict the occurrence of isolated spiking in CA1 at above chance levels, even in a window of 4 to 8 theta cycles before isolated spiking (Fig 7B). We also asked whether there was any evidence consistent with isolated spiking in CA1 influencing subsequent PFC activity (Fig 7C). We found that the coordination between the hippocampus and PFC persists after the occurrence of isolated activity but is weaker compared to intervals immediately before and during cycles with isolated activity (Fig 7D). We also found a weak but significant increase in prediction gain over the time bins, consistent with the strength of prediction increasing toward the isolated cycle ($R_{data}$ = 0.106, $p_{data}$ = 0.0073; $R_{perm.}$ = 0.0005, and $p_{perm.}$ = 0.990). We noted the average prediction gains were small in magnitude, which has previously been observed for prediction gains relating auditory and hippocampal activity around the times of SWRs [60]. This is not surprising given the relatively small numbers of simultaneously recorded PFC cells that were available to predict the activity of any given CA1 unit. We can therefore regard these cross-validated predictions as lower bounds on the actual values that would be obtained if it were possible to sample the entire PFC population. Indeed, examining the values for individual PFC ensemble—CA1 models revealed several cases with prediction gains between 2.5% and 5% (S8 Fig). Thus, our results demonstrate that

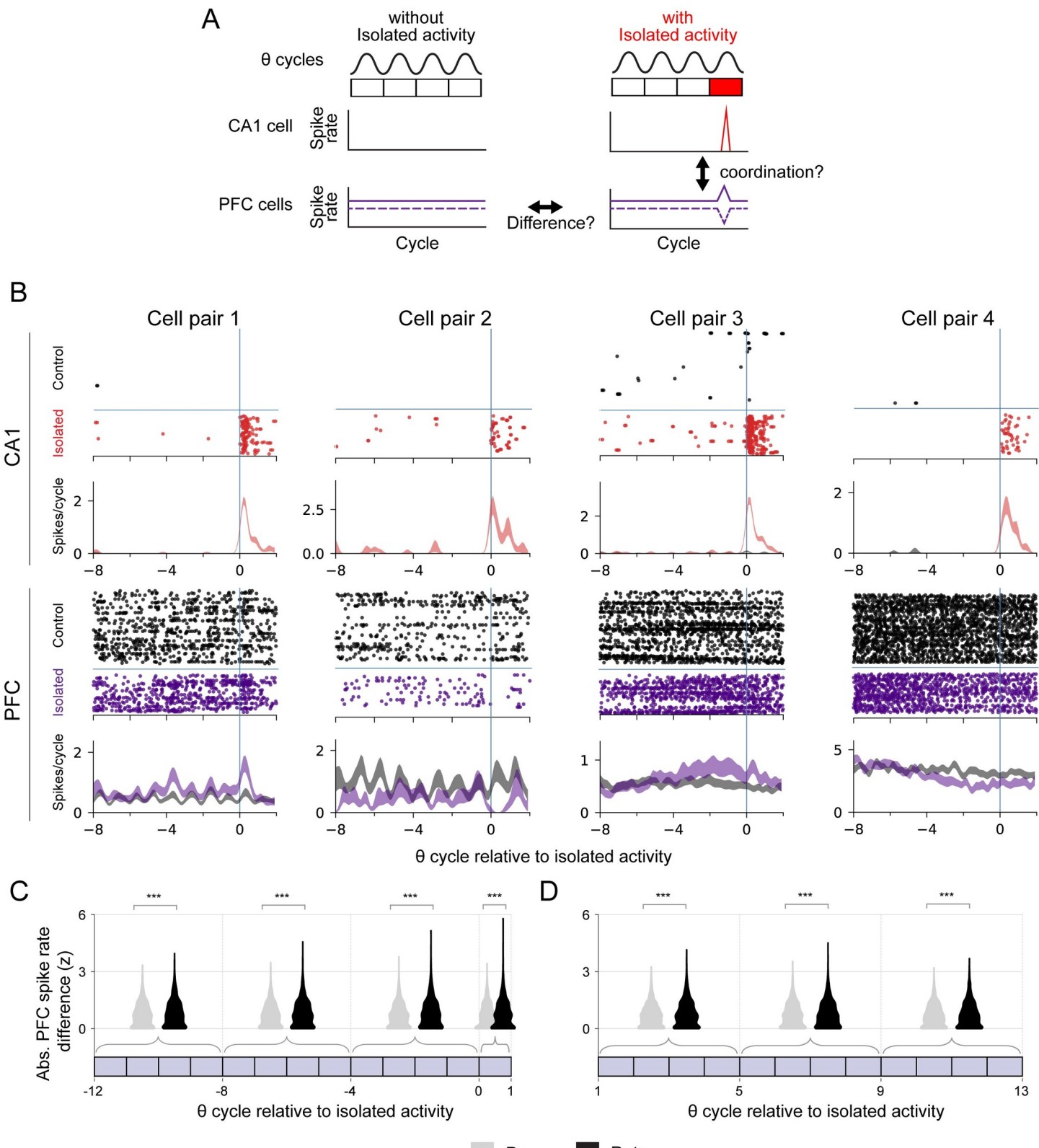

**Fig 6. PFC activity is coordinated with hippocampal isolated activity. (A)** Schematic illustrating potential CA1 and PFC activity around the time of isolated spiking. Changes in PFC spiking around the time of CA1 isolated spiking may reflect coordination between the 2 regions. **(B)** Example spike raster and spiking rate (mean ± SEM)

for pairs of co-recorded hippocampus CA1 and PFC cells. Each raster shows spiking aligned to isolated hippocampal activity (cycle 0) and matched control trials. Spiking is plotted relative to the cycles of the hippocampal theta rhythm. For CA1 cells, red indicates spikes and spiking rate for intervals with isolated spiking at cycle 0. Black indicates control intervals without isolated spiking at cycle 0. For PFC cells, purple indicates spikes and spiking rate for intervals with isolated spiking at cycle 0. Black indicates control intervals without isolated spiking at cycle 0. **(C)** Violin plots and quantification of spike rate differences between control and actual intervals for PFC–CA1 cell pairs ($n = 2{,}798$) in time windows relative to CA1 isolated activity. Rate difference, original data (black) and permuted (gray), is expressed the z-score of the absolute observed difference relative to its own permuted distribution. The Wilcoxon signed rank test (*** $p < 0.001$) was used to compare the original and permuted groups: $p = 4.7 \times 10^{-8}$, $3.6 \times 10^{-12}$, and $2.2 \times 10^{-10}$ for each group, respectively. **(D)** Violin plots and quantification of spike rate differences between control and actual intervals for PFC–CA1 cell pairs ($n = 2{,}892$) in time windows post CA1 isolated activity. The Wilcoxon signed rank test (*** $p < 0.001$, ** $p < 0.01$) was used to compare the original and permuted groups: $p = 9.5 \times 10^{-7}$, $1.8 \times 10^{-3}$, and $2.1 \times 10^{-3}$ for each group, respectively. PFC, prefrontal cortex.

information expressed by prefrontal cortical and hippocampal cell populations is coordinated around the time of isolated activity.

Importantly, the predictive PFC activity patterns were specific for individual CA1 cells. We examined the correlation between β coefficients of PFC predictors across predictive models. If the spiking of specific PFC cells was strongly predictive of isolated spiking of a particular CA1 cell but not of other CA1 cells, this β coefficient correlation should be low, indicating that a given PFC cell would predict the spiking in one model (e.g., one CA1 cell) but not another. By contrast, if a subset of PFC cells consistently predicted isolated spiking across CA1 cells, then

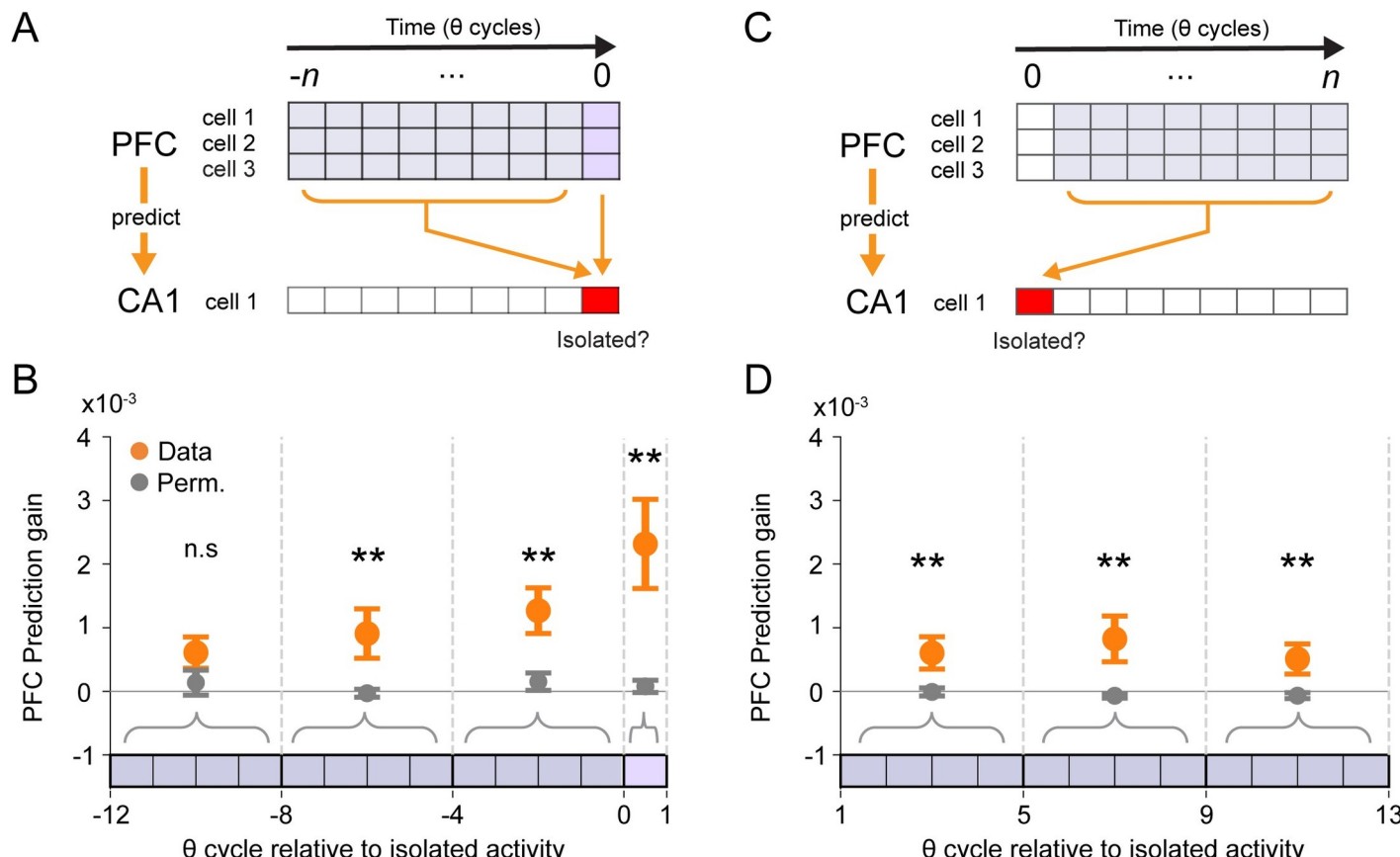

**Fig 7. PFC activity predicts the occurrence of hippocampal isolated spiking. (A)** PFC activity leading up to isolated spiking is used to predict the future occurrence of isolated spiking in one CA1 place cell. **(B)** Prediction gain (mean ± SEM) of GLMs where PFC spiking activity is used to predict isolated spiking in the upcoming CA1 theta cycle ($n = 158$). Pairwise permutation test (** $p < 0.05$) with multiple comparison correction: $p = 0.079$, $p = 0.0027$, $p = 0.0006$, and $p < 0.0002$ for each group, respectively. **(C)** PFC activity after isolated spiking is used to predict the previous occurrence of isolated spiking in one CA1 place cell. **(D)** Prediction gain (mean ± SEM) of GLMs where PFC spiking activity is used to predict whether CA1 isolated activity had occurred ($n = 162$). Pairwise permutation test (** $p < 0.05$) with multiple comparison correction: $p = 0.0071$, $p = 0.0002$, and $p = 0.0053$ for each group, respectively. GLM, generalized linear model; PFC, prefrontal cortex.

these correlations would be high, as the same PFC cells would show similarly β coefficients across models. We found that the mean correlation coefficient was not significantly different from 0 (median = −0.021, IQR = 0.16, Wilcoxon rank sum test $p$ = 0.431). This indicates that the PFC ensembles predicting the occurrence of isolated activity for different CA1 cells are distinct and argues for specificity in PFC–CA1 coordination around the occurrence of isolated activity.

## Discussion

We examined the isolated spiking seen outside of a place cell's place field with reference to local hippocampal network activity and prefrontal cortical activity. We found that this isolated spiking preferentially occurs during the late phase of theta oscillations, recapitulates coherent spatial representations, and is coordinated with prefrontal cortical activity. Our findings argue that seemingly spontaneous and sparse activity, previously considered as noise in the hippocampus, are precisely timed spikes that reflect coordinated activity both within the hippocampus and across hippocampal–prefrontal cortical networks.

We found evidence that CA1 isolated spiking reflects structured activity rather than noise within the hippocampal network at both the single cell and pairwise level of analysis. At the single cell level, isolated CA1 spiking is highly concentrated in the late phases of theta, pointing to the segregation of current versus noncurrent scenarios between the early and late phase of theta respectively [63]. This is in line with previously described place cell spiking associated with nonlocal representations including previously experienced environments [64], distinct spatial reference frames [65], possible future locations, travel in the noncurrent direction [66], and activity on nonpreferred trajectories [28], all of which are seen preferentially during the late phases of theta. Our results also extend previous findings of nonlocal spiking associated with vicarious trial and error behavior seen near choice points or at the edges of place fields [27,55]. We found that isolated spikes occur throughout the environment and are not concentrated at the edges of place fields or near choice points. These spikes also occurred in association with high movement speeds. They were more frequent during the early trials of a behavior session when the animal is attempting to find the new reward locations. Pairwise analyses further demonstrated that isolated spikes are coordinated across hippocampal neurons: Cells that fired together during adjacent spiking periods were also more likely to fire together within an isolated spiking event. This is consistent with a brief, coherent activation of a remote spatial representation, indicating that these events could support deliberative processes associated with the evaluation of distant physical locations.

Our analysis of isolated CA1 place cell spiking relative to PFC activity provided additional evidence that these events could reflect coordinated activity across brain regions. At the single cell level, we identified individual PFC neurons that spiked differently in association with isolated CA1 spiking events as compared to periods matched for location, direction of movement, and speed. At the ensemble level, we found that these differences were significant not only during the theta cycle associated with isolated CA1 spiking, but also for theta cycles around the isolated spiking event. Moreover, ensemble PFC activity could predict the occurrence of a theta cycle with isolated CA1 spiking, and these predictions remained significant for PFC activity occurring 4 to 8 cycles before the isolated spiking. These predictions were also specific: A particular set of PFC cells were strong predictors of a given CA1 cell's isolated spiking, while a different set of PFC cells might predict isolated spiking in a different CA1 cell.

These findings indicate that isolated spiking events in CA1 are very unlikely to be the product of local, stochastic fluctuations in the hippocampus. Instead, they are consistent with the transient expression of a "cell assembly" [51] whose activation represents the location where

the place fields of the constituent cells overlap. Our results indicate that specific patterns of PFC activity can predict the momentary expression of hippocampal cell assemblies. At the same time, the time course of this prediction (significant over approximately 1 second) implies a relatively slow PFC process and is not compatible with a simple fixed, distributed cell assembly that is engaged transiently and simultaneously across structures. Interestingly, the firing rate changes in PFC and the strength of the PFC–CA1 coupling is greatest during the theta cycle with isolated spiking but remains greater than expected by chance well after suggesting the possibility that the CA1 spiking drives a subsequent change in PFC activity. Thus, our findings point to a potential cortical–hippocampal–cortical information flow, conceptually similar to the cortical–hippocampal–cortical information flow seen around SWRs during sleep [60].

Our results also suggest that information coordination between cortex and hippocampus may occur frequently during active behavior. This extends previous findings of hippocampal–prefrontal coupling from imaging [67–73] and neural recording experiments [37–44,74–77] on the role of PFC in modulating both cortical and subcortical structures during mnemonic processes [35,73,74,78,79]. Our results also complement findings demonstrating coherent spiking activity patterns across the hippocampus and PFC in the context of both SWRs and locomotion-associated spiking [37,59,77,80].

Interestingly, the communication latency between PFC and the hippocampus is hypothesized to be approximately 26 to 28 ms or approximately ¼ of a theta cycle [43]. We were therefore surprised to find PFC spiking can predict whether isolated spiking will occur up to 4 to 8 theta cycles or approximately 500 ms to 1 second earlier, an interval much longer than what is needed for the hypothesized direct information transfer. Although the channel for communication between PFC and the hippocampus may have a short latency, our results suggest the expression of isolated hippocampal spiking likely involves coordinated activity between these regions [67–69] that evolves over time [70–72]. This is consistent with human imaging studies that show cortical activity change can precede memory recall on the order of seconds [70]. The long timescale is also consistent previous observations of prefrontal–hippocampal coordination overt deliberation at choice points [77]. This long duration may in part be explained by the timescale of cortical processing where spiking time constants are >100 ms [81]. The long timescale may reflect additional intracortical communication necessary to integrate information across multiple theta cycles, which eventually triggers the expression of hippocampal representations.

Previous findings indicate activity in the hippocampus can switch between current or local to noncurrent or nonlocal representations on a fast scale [28,64,65]. Our results represent a distinct mode of such hippocampal function. Our data point to a role for PFC in potentially modulating isolated spiking in the hippocampus. However, further work is necessary to determine whether other nonlocal hippocampal representations are also modulated by PFC.

Given the higher frequency of isolated spiking during early trials in a behavior session, when the animal is actively identifying new reward locations, we hypothesize that these coordinated events may serve to modulate ongoing hippocampal–cortical network representations, corresponding to current experience, with internally generated representations, corresponding to noncurrent scenarios. Additionally, we hypothesize that through this mode of coordination, PFC could drive the expression of noncurrent scenario representations from memory in the hippocampus, which, in turn, feed back to cortical regions as a part of an evaluation loop [82]. This coordination could be involved in the covert evaluation of potential trajectories or goal locations for decision-making in the future. The cortical drive could potentially underlie previously reported extra-place field spiking and noncurrent spatial representations in the hippocampus associated with approach to a choice point [28], during vicarious trial and error [27], and spiking during travel in the nonpreferred direction of place fields [6–9,28].

Both cortical and hippocampal spiking patterns display noise-like variation, even when the animal performs repeated tasks or actions [83–85]. However, cortical discharge can also be highly reproducible given a consistent input [86], and behaviors can reflect a degree of accuracy consistent with very low levels of noise in the brain [87]. In the context of signal versus noise, our findings indicate that the sparse spiking of hippocampal place cells is better understood as signal, potentially containing noncurrent representations consistent with alternative scenarios. These transient injections of noncurrent representations could signal processes that update ongoing hippocampal representations. Thus, the hippocampal place cell spiking during active behavior may dynamically reflect both externally driven and internally generated, noncurrent representations, which we hypothesize can collectively guide ongoing behavior.

## Methods

The data used in this study came from the same data set used in previous publications [45,46].

### Ethics statement

Experiments followed guidelines from the University of California San Francisco Institutional Animal Care and Use Committee (protocol #AN174991) and US National Institutes of Health. The University of California San Francisco Institutional Animal Care and Use Committee meets all applicable federal, state, and university laws and regulations. USDA (#9199: #93-R-0440) and California Department of Fish and Wildlife (#1736).

### Animal and behavior

Six Long–Evans rats (male, 500 to 700 g, 4 to 9 months of age) were first trained to traverse a linear track (1 m) for reward (evaporated milk, Carnation brand, with 5% added sucrose). Next, the animals were trained on a foraging task approximately 21 days after surgery [45,46]. Briefly, the task has 4 possible reward well locations; only 2 were chosen to deliver reward at a given time. The rat is trained to visit the 2 rewarded locations in alternation to receive reward. The rewarded well locations changed within or between sessions or between days. Moreover, 2 to 3 sessions (15 to 45 minutes) were performed each day with interleaved rest sessions (20 to 60 minutes). Reward was delivered (100 to 300 μl at 20 ml/min) using a syringe pump (NE-500 OEM, New Era Pump Systems, Farmingdale, NY, USA.) after the animal broke an infrared beam at the well location.

### Implant

The recording drive was 3D printed (PolyJetHD Blue, Stratasys, Eden Prairie, MN, USA.) and contained up to 28 individually movable tetrodes. Tetrodes (Ni-Cr, California Fine Wire, Grover Beach, CA, USA.) were gold plated to 250kOhm at 1 kHz.

The implanted recording drives targeted both dorsal CA1 (7 tetrodes) and dorsal PFC (14 to 21 tetrodes, housed in one cannula angled at 20 degrees toward the midline): CA1 Anterior-Posterior (AP): −3.8 mm and Medial-Lateral (ML): 2.2 mm. PFC (anterior cingulate cortex and dorsal prelimbic cortex): AP: +2.2 mm, ML +1.5 mm and Dorsal-Ventral (DV) between 1.88 mm and 2.72 mm depending on the AP and ML coordinates of each tetrode.

Initially, tetrodes were adjusted to reach the target DV coordinate (PFC) or guided by LFP and spiking patterns (CA1) every 2 days. Once the target was reached, tetrodes were adjusted (approximately 30 μm) to improve cell isolation at the end of an experiment day.

## Histology

Recording sites were marked with electrolytic lesions by passing current through each tetrode (30 μA, 3 seconds) at the end of the experiment. Animals were perfused after 24 hours with paraformaldehyde (4% in PBS). The brain was removed, fixed (24 hours at room temperature), cryoprotected (30% sucrose in PBS at 4˚C), and sectioned (coronal, 50 μm). Cresyl violet was used to stain the sections to identify sites of electrolytic lesions.

## Recording

Data were recorded with the NSpike system (LMF and J. MacArthur, Harvard Instrumentation Design Laboratory, Cambridge, MA, USA.). Dim lighting was used during the experiment. An infrared LED array was mounted on the headstage amplifier to for position tracking. Video was recorded at 30 Hz. We recorded LFP (0.5 to 400 Hz at 1.5 kHz) and spiking activity (600 to 6,000 Hz or 300 to 6,000 Hz at 30 kHz) from each tetrode channel. For spike detection referencing, a tetrode located in corpus callosum was used for CA1, and a local tetrode without detected spikes was used for PFC.

## Data preprocessing

Manual spike clustering was performed based on peak amplitude, spike width, and waveform principal components (MatClust, https://bitbucket.org/mkarlsso/matclust/src/master/).

To reconstruct the position of the animal, the centroid of the front and back diodes from the LED array was automatically extracted from the video.

## Spike clustering quality

To assess clustering quality, we analyzed the similarity in spike waveform within and across different units. We expect a well-clustered unit to have spikes with waveforms that are similar to other spikes assigned to the same unit compared with spikes assigned to other units. Potential spike misassignment can occur only for spiking events detected on the same tetrode. We therefore compared spike waveforms from a given unit to spikes from all other units on the same tetrode. We computed the Euclidean distance between the spike waveforms (4 channels) for all pairs of spikes. Next, we compared the minimum waveform distance between spikes belonging to the same unit and between that unit and all other units. This was done separately for spikes associated with isolated or adjacent theta cycles.

## Spatial spiking rate

The occupancy normalized rate was calculated by dividing the number of spikes by the occupancy of the animal per spatial bin (2 cm by 2 cm) in the environment. A 2D symmetric Gaussian kernel (σ = 2-cm and 12-cm spatial extent) was then used for smoothing.

## Theta cycle definition and classification

The theta frequency component of the raw LFP signal was extracted using an equiripple finite impulse response band-pass filter (6 to 12 Hz). Given theta is associated with movement states [48], we used 2 criteria to exclude activity associated with immobility periods. First, we exclude periods when the speed of the animal was less than 2 cm/s. Second, we excluded periods with SWRs, which occur during immobility or periods of slow movement, using previously described methods [45,46]. For SWR detection, we used a speed threshold of <4 cm/s to ensure that SWRs occurring during intermediate speeds (>2 cm/s but <4 cm/s) are excluded. Spikes occurring during these excluded periods are classed as "Excluded."

For spiking during the included periods, we classified each theta cycle and spikes belonging to that cycle as "isolated" or "adjacent" activity. This was done per place cell. The classification was based on the mean number of cycles separating a given cycle with spiking to its nearest 3 other neighboring cycles with spiking. A mean separation of 8 cycles is the threshold for classification as adjacent as opposed to isolated. This is based on the distribution of cycle separation across the entire place cell population.

### Theta cycle spatial distribution

To determine if isolated spiking occurred more frequently at certain locations in the environment compared with adjacent spiking, we first plotted the normalized spatial distribution of theta cycles containing each type of activity for each cell, averaged across the population. The spatial distribution of isolated and adjacent theta cycles for each place cell was calculated using the spatial spiking rate method described above, except with their respective cycles instead of spikes. To determine whether there are areas in the environment where the occurrence of adjacent and isolated activity differ, we applied a permutation technique. This involves first permuting the identity of each theta cycle labeled as having adjacent or isolated activity. The spatial distribution of the 2 permuted sets were calculated and subtracted from each other to obtain the difference. This was done 500 times to generate an expected distribution of differences. The actual difference in spatial distribution between isolated and adjacent activity was compared to the expected distribution and a z-score was calculated.

### Theta cycle movement correlates

To determine whether isolated and adjacent activity were associated with distinct movement correlates, we compared the distribution of animal speed and angular acceleration at times of theta cycles containing each type of activity. This was done for each place cell and then averaged across the population. To determine if the 2 distributions were significantly different, we used a permutation approach. For each place cell, the identity of the theta cycle, whether it contains isolated or adjacent, was permuted. The difference between the 2 distributions was then recalculated. This was repeated 1,000 times to obtain a distribution of expected differences. The actual difference was expressed as a z-score relative to the expected distribution.

### Theta phase locking analysis

The theta phase of each spike is relative to the phase of the reference signal obtained from the tetrode located in the corpus callosum. The mean phase preference for spiking activity for each place cell is the circular mean of the phases of all spikes. Theta phase concentration is the magnitude of the vector sum of all spikes, where each spike is a unit vector with angle corresponding to the phase of theta. We limited the analysis to cells with at least 50 adjacent and 5 isolated spikes.

### Task performance

We computed the performance curve of the animal using a state-space algorithm [88,89]. The performance is probability of the animal visiting the pair of rewarded locations for a given contingency.

### Interspike distance, time, and angle analysis

We calculated the distance between the locations of one spike and every other spike using Dijkstra algorithm [90], which returns the shortest linear distance accounting for the topology

and geometry of the maze. To generate the distribution of distance between spike pairs, we calculated the mean distance between one spike and its nearest 5 spikes.

To compute the time elapsed between spikes that occurred at similar locations, we selected one reference spike and all other spikes of interest that occurred within a 1-cm radius. We then calculated the pairwise difference in time between the reference spike and all other selected spikes and took the mean.

To compute the difference in the direction of travel between spikes that occurred at similar locations, we followed the same approach by finding spike pairs occurring within a 1-cm radius. We then constructed a vector spanning the locations of the animal in a 1-second window centered on the spike to represent the direction of travel for each spike. Next, we calculated the mean absolute difference in angle between the reference spike vector and all other spike vectors.

For each cell, we generated a distribution across the range of values for distance, time, and angle. We then averaged the distribution across cells with at least 50 adjacent and 5 isolated spikes.

## Time aligned spectrogram

To compute the spectral properties of network activity around spiking events we first used a bank of band-pass filters (center frequency ± 1Hz) to filter the LFP signal across the frequency range 2 to 250Hz. Each filtered signal normalized by subtracting the mean and dividing by the standard deviation. For each place cell, we then selected the normalized signal in a 500-ms window centered on the time of each spiking event and averaged across spikes. This was repeated for spikes classified as belonging to isolated, adjacent theta cycles or excluded from analysis (see "Theta cycle definition and classification"). We ensured equal numbers of spikes were used to generate the average across spike types for each cell by sampling without replacement to match the type with the lowest count. The average for each place cell was then used to generate the mean for the entire population.

## Spiking coactivity

We quantified the likelihood of a pair of place cells having isolated activity in the same theta cycle relative to the expected probability, similar to what has been done for SWRs [45,58,91]. The expected probability is the frequency of observing spiking from 2 cells in the same theta cycle given their relative frequency of spiking. For each cell, its spike count during a theta cycle with isolated activity was first binarized, where the cell was either spiking or not spiking in that theta cycle. The proportion of all theta cycles where both cells spiked was the observed coactivity. The expected coactivity was calculated by permuting the participation of each cell across all theta cycles with isolated activity. This was repeated 1,000 times and to generate a distribution expected proportion of theta cycles with both cells have isolate activity. The observed proportion was converted to a z-score by subtracting the mean and dividing by the standard deviation of the expected distribution. This method accounts for the differences in the number of theta cycles with isolated activity for each cell in the pair.

To determine the temporal relationship between adjacent activity for a pair of place cells, we computed the cross-correlation between theta cycles with adjacent activity for a given cell pair. First, we assigned each theta cycle of a given cell as having adjacent spiking or not. Then, we cross-correlated the assignment for a pair of place cells, where the lag is measured in the number of cycles. We then found the absolute lag with the maximum cross-correlation value with for each place cell pair.

## Cycle matching

For each place cell, we matched each theta cycle with isolated activity with control theta cycles without spiking. These control cycles were drawn from other task trials and matched as closely as possible for trajectory, speed, and location. Two control cycles were selected for each actual cycle. Trajectory matching only included task trials where the animal performed the same trajectory and ensured the same direction of travel across all matched cycles. The speed matching process started with generating a reference speed profile distribution for a time interval around a theta cycle with isolated activity for a given cell. For each theta cycle with isolated activity, we then chose 2 candidate theta cycles without spiking. The speed profile for each candidate cycle around the same interval was compared with the reference distribution. The candidate cycle was accepted if the mean speed deviation compared with the reference distribution is $<1\sigma$. The next inclusion criteria for the candidate cycle was having a location $<10$ cm from the theta cycle with isolated activity. This selection process was done without replacement. Only place cells with greater than 100 input cycles, including both isolated and matched cycles, were included in the analysis.

## Spiking normalization to theta cycles

For illustration purposes in Fig 6B, we converted PFC and CA1 spiking times to hippocampal theta cycle phases. Spiking times were transformed using linear interpolation from time to theta phase relative to the start of the theta cycle with isolated activity. The mean spiking rate was calculated with respect to theta cycles.

## Spiking rate comparison

We asked how PFC spiking rate leading up to and including theta cycle with isolated activity differed from matched control cycles. For each CA1 cell, we first identified cycles with isolated spiking and control cycles without isolated spiking (see above). We next found all PFC cells that were simultaneously recoded with the CA1 cell. For each of these PFC cells, we compared the spike count in time intervals leading up to the theta cycle with or without isolated spiking from the CA1 cell. Under the null hypothesis, the difference between the 2 sets of spike counts will not be significantly different than chance. To estimate the significance of the spike count difference, we used a permutation test where we permuted the theta cycle identity 1,000 times and calculated the difference between the PFC spiking for each permutation. The actual difference was expressed as a z-score relative to this permuted distribution by subtracting the mean and dividing by the standard deviation of the permuted distribution. As an additional control to estimate the expected difference between the groups, we repeated the analysis by first generating a permuted data set where the theta cycle identity (with or without isolated spiking) was permuted. This difference in the spiking rate of this permuted data set, expressed as a z-score, was calculated as per the actual data set.

## Generalized linear models

We asked whether spiking activity from simultaneously recorded PFC cells can predict the occurrence of isolated activity from a CA1 cell. We built cross-validated generalized linear models (GLMs) (binomial distribution with logit link function) with elastic net regularization, which combined LASSO and Ridge regularization to reduce overfitting [92]. To do this, we first identified theta cycles with isolated spiking for a CA1 cell. We next identified another control set of theta cycles when the CA1 cell did not spike. These control cycles were matched for animal speed, movement direction and location (see "Cycle matching"). We created a model

for each CA1 cell to determine whether PFC spiking activity can distinguish between cycles with or without isolated spiking in a time window relative to the cycle with isolated spiking. We first made models using activity in the 12 cycles previous to the cycle with isolated activity and then the 12 cycles after the cycle with isolated activity. We ensured that no other isolated activity that occurred in this window was used for prediction. A 4-cycle bin size was used for grouping PFC activity since PFC activity shows relatively long autocorrelation times.

## Modeling parameters

MATLAB's *lassoglm* function was used ("distr" = "binomial," "Link" = "logit"). The optimization was equally weighed between LASSO and Ridge methods ("alpha" = 0.5). Shrinkage parameter ($\lambda$) optimization was done using 3-fold cross-validation ("CV" = 3) with 5 Monte Carlo repetitions ("MCReps" = 5). We used 5-fold cross-validation and averaged the outcome across the 5 cross-validations.

## Prediction gain

Prediction gain describes whether the models can predict the outcome above chance. For the actual data set (Prediction gain$_{Data}$), we did this by first calculating the mean absolute error (MAE) between the predicted and actual outcome for the validation partition (MAE$_{Data}$). To estimate chance performance, we repeated the prediction 5,000 times, each time with the outcome permuted, and calculated the MAE. The chance MAE is the mean MAE of the 5,000 control predictions (MAE$_{Permuted}$). The prediction gain is $\log_{10}($MAE$_{Permuted}/$MAE$_{Data})$. A positive prediction gain means the Data group had a smaller error, or better prediction, compared with the permutated group [60].

We also used a second approach to estimate chance prediction. Instead of building the model using actual data, we permuted the trial identity of the PFC input, which preserves the input spiking distribution but destroys any potential relationships between trials and the outcome in CA1. We repeated the entire modeling procedure using permuted data and calculated the prediction gain (Prediction gain$_{Permuted}$).

To estimate whether there is significant above chance prediction of CA1 isolated activity from PFC activity, we performed a permutation test ($n$ = 10,000 permutations) on the mean prediction gain between the actual (Prediction gain$_{Data}$) and permuted (Prediction gain$_{Permuted}$) data sets. The Bonferroni correction was used to adjust the significance of the prediction to account for multiple comparisons between time windows.

## PFC predictive ensemble correlation

To determine whether there is specificity in the coordination between PFC and CA1 around the time of isolated activity, we asked if isolated activity for each CA1 cell was predicted by activity from distinct combinations of PFC cells. This was done by calculating the Pearson correlation between the $\beta$ coefficients of CA1 models that were generated from data recorded on the same day. We selected models that yielded a prediction gain >1, had a minimum of 5 predictors, and had at least 2 CA1 models from the same day. This produced a data set from 17 days, with a median of 3 (Q1: 2, Q3: 5) models per day, with 21 (Q1: 19, Q3: 23) predictors per model, and with a median prediction gain of 1.014 (Q1: 1.0015, Q3: 1.026).

## Model quality assessment

We checked the quality of our models by examining the relationship between prediction gain and the contribution of predictors to the model. For these linear models, we used the value of

the β coefficient to indicate the contribution of a predictor to the prediction, where predictors with nonzero β coefficient may contribute to the prediction. We examined how the prediction value varied with the proportion of predictors with nonzero β coefficients, or the total number of input predictors, using linear regression. We also compared these relationships between models with actual data or permuted data. Models with greater predictive power are expected to have higher proportions of predictive features whereas the number of input predictors should not affect the outcome.

## Statistical analyses

Circular statistical analyses were performed using the Circular Statistics Toolbox in MATLAB [92]. Statistical tests were performed using standard MATLAB modules, Statsmodels Statistical Functions, and Scipy Statistical Functions (scipy.stats). All tests were 2 sided.

## Supporting information

**S1 Fig. Adjacent and Isolated activity. (A)** Classification of adjacent versus isolated activity based on temporal separation between theta cycles with spiking. **(B)** Proportion of spikes (adjacent and isolated) classified as isolated for all CA1 cells. Median: $0.17 \pm 0.018$ (95% CI).
(TIF)

**S2 Fig. Spatial distribution of adjacent spiking is not correlated with the proportion of isolated spiking. (A)** Distribution of spatial information for CA1 cells ($n = 247$). Median: 4.35 bits/spike. **(B)** The proportion of isolated spikes is very weakly correlated with spiking spatial information ($n = 247$). $R^2 = 0.0246$ $p = 0.014$. **(C)** Distribution of the median distance between adjacent spikes ($n = 247$). Median: 24.04 cm. **(D)** The proportion of isolated spikes is not significantly correlated with the median distance between locations at which adjacent spikes were observed ($n = 247$). $R^2 = 0.0081$ $p = 0.16$. **(E)** Distribution of the standard deviation of distances between locations at which adjacent spikes were observed ($n = 247$). Median: 40.18 cm. **(F)** The proportion of isolated spikes is not significantly correlated with the standard deviation of distances between locations at which adjacent spikes were observed ($n = 247$). $R^2 = 0.0089$ $p = 0.14$.
(TIF)

**S3 Fig. Isolated spiking activity is not due to spike assignment errors. (A)** Minimum Euclidean distance between the spike waveform of adjacent activity spikes within each cell versus all other cells recorded on the same tetrode ($n = 260$). Wilcoxon signed rank test: $p = 2.9 \times 10^{-51}$. **(B)** Minimum Euclidean distance between the spike waveforms of spikes classified as isolated activity within each cell versus all other cells recorded on the same tetrode ($n = 276$). Wilcoxon signed rank test: $p = 8.8 \times 10^{-51}$.
(TIF)

**S4 Fig. Spiking phase concentration and phase preference for isolated spikes by distance from adjacent spiking. (A)** Mean theta phase preference distribution for isolated spiking grouped by distance to its nearest 10 adjacent spikes. Mean phase: 0.80, 0.97, 1.13, and 0.89 for distances 0–2, 2–8, 8–16, and $\geq$16 cm, respectively. Only cells with 5 or more spikes are included for each distance category. Kruskal–Wallis test: H (3, 815) = 9.79, $p = 0.020$. **(B)** Mean spiking phase concentration for isolated spiking grouped by distance to its nearest 10 adjacent spikes. Median phase concentration: 0.39, 0.46, 0.49, and 0.34 for distances 0–2, 2–8, 8–16, and $\geq$16 cm, respectively. Only cells with 5 or more spikes are included for each distance category. Kruskal–Wallis test: H (3, 815) = 52.8, $p = 2.03 \times 10^{-11}$.
(TIF)

**S5 Fig. Hippocampal network spectral signatures around excluded, adjacent and isolated spiking activity. (A)** Mean spike triggered spectrogram for excluded (left), adjacent (center), and isolated (right) spiking activity ($n$ = 170 cells). Top panels show frequency ranges 50 to 250 Hz. Bottom panels show frequency ranges 2 to 50 Hz. **(B)** Mean spectral power for a 50-ms window centered at 0-ms lag (median ± 95% CI).
(TIF)

**S6 Fig. Each theta cycle with isolated place cell activity is matched with nonisolated cycles for speed, trajectory, and location. (A)** Speed match profiles for examples in Fig 6B. **(B)** Distribution of mean difference in speed between matched and isolated cycles. Speed profiles of matched cycles were on average within −0.06 standard deviations of the speed profile of the isolated cycles. The difference is expressed as a z-score normalized against the speed distribution of isolated cycles. **(C)** Location match profiles for examples in Fig 6B. **(D)** Distribution of the mean distance in cm between matched and isolated cycles. The location of the animal on matched cycles was on average 7.5 cm from the location of the isolated cycle. **(E)** Intercycle time interval between isolated cycles or isolated and matched cycles for CA1 cells with at least 100 cycles ($n$ = 158). Sign test $p$ = 0.0669.
(TIF)

**S7 Fig. PFC firing rate change in time bins relative to isolated CA1 spiking. (A)** Distribution of PFC firing rate change (z) for time windows before and during CA1 isolated spiking in Fig 6C. Bins above and below 2 z are colored red and the proportion of the distribution are printed in red. Only PFC–CA1 cell pairs where PFC spiking occurs in at least 25% of isolated or matched cycles are included. This selects for PFC cells with higher spiking rates. **(B)** Distribution of permuted PFC firing rate change (z) for time windows before and during CA1 isolated spiking in Fig 6C. PFC, prefrontal cortex.
(TIF)

**S8 Fig. Model quality controls for GLMs using PFC activity to predict CA1 isolated activity. (A)** Input predictor count for actual and permuted data sets. Wilcoxon rank sum test $p$ = 1.0. **(B)** Prediction gain is not significantly correlated with the total number of input predictors used for prediction. Data: $R^2$ = 0.00121, $p$ = 0.384; permutation control: $R^2$ = 0.00387, $p$ = 0.118. **(C)** Models using actual data have higher proportions of predictors with nonzero β coefficients. Wilcoxon rank sum test $p$ = 1.22 × 10$^{-16}$. **(D)** Prediction gain is positively correlated with the proportion of input predictors with nonzero beta coefficients. This is found in both actual (left) and permuted (right) data sets. Data: $R^2$ = 0.133, $p$ = 3.10 × 10$^{-21}$; permutation control: $R^2$ = 0.0537, $p$ = 0.3.80 × 10$^{-9}$. Each point in the scatter represents a single fold of each model with 5 folds in total. All time points and models are shown. GLM, generalized linear model; PFC, prefrontal cortex.
(TIF)

**S1 Data. Data for Fig 1E.** S2 Data. Data for Fig 1F. Values are in log$_{10}$. S3 Data. Data for Fig 2B. S4 Data. Data for Fig 2C. S5 Data. Data for Fig 2D. S6 Data. Data for Fig 3B. S7 Data. Data for Fig 3C. S8 Data. Data for Fig 4B. S9 Data. Data for Fig 4C. S10 Data. Data for Fig 4D. S11 Data. Data for Fig 4E. S12 Data. Data for Fig 4F. S13 Data. Data for Fig 4G. S14 Data. Data for Fig 5B. S15 Data. Data for Fig 6C. NaN values are excluded. S16 Data. Data for Fig 6D. NaN values are excluded. S17 Data. Data for Fig 7B. S18 Data. Data for Fig 7D. S19 Data. Data for S1B Fig. S20 Data. Data for S2A Fig. S21 Data. Data for S2B Fig. S22 Data. Data for S2C Fig. S23 Data. Data for S2D Fig. S24 Data. Data for S2E Fig. S25 Data. Data for S2F Fig. S26 Data. Data for S3A Fig. S27 Data. Data for S3B Fig. S28 Data. Data for S4A Fig. S29 Data. Data for

S4B Fig. S30 Data. Data for S5B Fig. S31 Data. Data for S6A Fig panel 1. S32 Data. Data for S6A Fig panel 2. S33 Data. Data for S6A Fig panel 3. S34 Data. Data for S6A Fig panel 4. S35 Data. Data for S6B Fig. S36 Data. Data for S6D Fig. S37 Data. Data for S6E Fig. S38 Data. Data for S7A Fig. S39 Data. Data for S7B Fig. S40 Data. Data for S8A Fig. S41 Data. Data for S8B Fig panel 1. S42 Data. Data for S8B Fig panel 2. S43 Data. Data for S8C Fig. S44 Data. Data for S8D Fig panel 1. S45 Data. Data for S8D Fig panel 2.
(ZIP)

## Author Contributions

**Conceptualization:** Jai Y. Yu, Loren M. Frank.

**Data curation:** Jai Y. Yu.

**Formal analysis:** Jai Y. Yu.

**Investigation:** Jai Y. Yu.

**Methodology:** Jai Y. Yu.

**Project administration:** Jai Y. Yu.

**Resources:** Loren M. Frank.

**Supervision:** Jai Y. Yu, Loren M. Frank.

**Validation:** Jai Y. Yu.

**Visualization:** Jai Y. Yu.

**Writing – original draft:** Jai Y. Yu, Loren M. Frank.

**Writing – review & editing:** Jai Y. Yu, Loren M. Frank.

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
