## [Editor Report · Decision Letter 0]

1 Jun 2021

Dear Dr Yu, 

Thank you for submitting your manuscript entitled "Prefrontal cortical activity predicts the extra-place field spiking of hippocampal place cells" for consideration as a Research Article by PLOS Biology.

Your manuscript has now been evaluated by the PLOS Biology editorial staff, as well as by an academic editor with relevant expertise, and I am writing to let you know that we would like to send your submission out for external peer review. Please accept my apologies for the delay in sending this decision to you.

Please re-submit your manuscript within two working days, i.e. by Jun 03 2021 11:59PM.

Kind regards,

Gabriel Gasque

Senior Editor

PLOS Biology

ggasque@plos.org

---

## [Decision Letter · Decision Letter 1]

30 Jun 2021

Dear Dr Yu,

Thank you very much for submitting your manuscript "Prefrontal cortical activity predicts the extra-place field spiking of hippocampal place cells" for consideration as a Research Article at PLOS Biology. Your manuscript has been evaluated by the PLOS Biology editors, by an Academic Editor with relevant expertise, and by three independent reviewers.

In light of the reviews (below), we are very positive about your manuscript and pleased to offer you the opportunity to address the comments from the reviewers in a revised version that we anticipate should not take you very long. We will then assess your revised manuscript and your response to the reviewers' comments and we may consult the reviewers again.

**IMPORTANT: While addressing the comments from the reviewers, please also address the editorial requests listed below my signature. 

We note that in the submission system you stated, “All data are fully available without restriction. Data used for this manuscript can be accessed at: https://crcns.org/data-sets/hc/hc-13/about-hc-13”. However, on the website, you state, “Conditions for using the data. These data required years of effort to collect, and it is the lab policy that anything published using these data list the person who collected the data and Dr. Frank as authors."

Unfortunately, this is a restriction that is not complaint with our Data Availability Policy (https://journals.plos.org/plosbiology/s/data-availability). Our data is published under a CC-BY license that allows unrestricted re-use of the data as long as the original contribution is credited. Thus, for PLOS Biology to continue consideration of your manuscript, we would need you to remove that restriction

It seems there is more data at https://crcns.org/data-sets/hc/hc-13/about-hc-13 then strictly what underlies this study, and so another possibility would be to deposit the relevant data used in this study separately and make that available without restriction. We would strongly endorse either of these options of making you data available.

At a minimum, you would need to unrestrictedly share all individual quantitative observations that underlie the data summarized in the figures and results of your paper. For an example see here: http://www.plosbiology.org/article/info%3Adoi%2F10.1371%2Fjournal.pbio.1001908#s5

These data can be made available in one of the following forms:

1) By updating your current deposition in crcns.org or depositing the data underlying this study separately and without restriction. In either case, please provide the accession code or a reviewer link so that we may view your data.

2) By providing Supplementary files (e.g., excel).

You can find more information on our data policy requests below my signature. Please feel free to contact me by email if you have any questions (lsmith@plos.org). 

We expect to receive your revised manuscript within 1 month.

**IMPORTANT - SUBMITTING YOUR REVISION**

*Resubmission Checklist*

*Published Peer Review*

*Blot and Gel Data Policy*

Sincerely,

Lucas Smith (on behalf of Gabriel Gasque)

Associate Editor

PLOS Biology

lsmith@plos.org

ETHICS REQUESTS

Please include the ID number of your experimental protocols approved by the University of California San Francisco Institutional Animal Care and Use Committee.

Please include the specific national or international regulations/guidelines to which your animal care and use protocol adhered. Please note that institutional or accreditation organization guidelines (such as AAALAC) do not meet this requirement.

DATA SHARING REQUEST

As discussed above PLOS has a data policy, which requires that all data be made available without restriction: http://journals.plos.org/plosbiology/s/data-availability. For more information, please also see this editorial: http://dx.doi.org/10.1371/journal.pbio.1001797

Note that we do not require all raw data. Rather, we ask for all individual quantitative observations that underlie the data summarized in the figures and results of your paper. For an example see here: http://www.plosbiology.org/article/info%3Adoi%2F10.1371%2Fjournal.pbio.1001908#s5

These data can be made available in one of the following forms:

Regardless of the method selected, please ensure that you provide the individual numerical values that underlie the summary data displayed in the following figure panels: Figures 1EF, 2B-D, 3BC, 4B, 5C, 6BD, S1, S2, S3AB, S4B-E, S5A-E, and S6BD.

***Please also ensure that each figure legend in your manuscript includes information on where the underlying data can be found and that your supplemental data file/s has/have a legend.

***Please ensure that your Data Statement in the submission system accurately describes where your data can be found.

REVIEWS:

Reviewer #1, Freyja Ólafsdóttir: Yu and colleagues aim to elucidate the function of place cell out-of-field activity. Traditionally, spikes occurring outside a place cell's (or any other neuron's) spatial receptive field is considered as noise. However, if this sporadic activity really reflects noise one would expect it to be uncoordinated in relation to other local and non-local cell activity. Yu and Frank find this is not the case. Rather out-of-field place cell activity (termed 'isolated spikes' by the authors) shows a strong phase preference along the movement-related theta rhythm, notably the phase preference differs for the isolated spikes compared to receptive-field spikes (termed 'adjacent spikes'). Further, adjacent co-activity patterns reflecting spatial field overlap are reactivated during isolated spikes. Finally, the study also shows that CA1 isolated spiking correlates with activity in the PFC and may be predicted by PFC activity. 

 Overall, I find the study's results interesting and I think the results may shed light on an under-studied activity phenomenon. I do have some questions regarding *what* this isolated activity reflects and require some methodological clarification regarding CA1-PFC coordination analyses. I outline my concerns below. All suggestions should be easily addressable by the authors and does not require the collection of more data. 

 1. My main questions concerns what this out-of-field activity represents. The authors suggest isolated activity could serve a sort of planning-like function serving decision making and goal directed behaviour. However, it would be useful to understand if isolated activity spikes could also be explained by other parameters. 

 a. For example, the authors show some isolated spikes seem to occur at locations where a place cell has a spatial receptive field, but for a different running direction. Could the authors show what proportion of isolated activity could perhaps be explained by such incomplete directional remapping?

 b. Moreover, the animals are carrying out a highly dynamic task on the maze - i.e. the reward locations change every day/session. Could perhaps the isolated activity reflect the remnants of previous place fields or more generally instability of spatial representations on the task? Although this point is a bit tricky to answer, the authors could check if the occurrence of isolated spikes differs for cells that have ore more less spatial representations on the maze. Relatedly, if there are days/sessions where the reward contingency did not change for a protracted period, the authors could assess if in the more stable task conditions that occurrence of isolated spikes is reduced. 

 This points speaks partly to recent findings showing artificial remapping in CA1 cells is not random but can rather be predicted by residual (i.e. isolated) activity prior to the induction of remapping, suggesting the existence of 'dormant' place fields. Thus, could the isolated activity the authors describe here reflect such dormant firing fields?

 2. Fig 5B shows examples of isolated and control activity from CA1 cells, however I am unsure how these plots are made. Does each row represent a trial (or a run) through a particular location? Is the location the same across trials, or is each row showing an example of isolated activity occurring in different places? Moreover, some of these examples show clear clustering of CA1 spikes around cycle 0, how can this be? The definition of isolated activity is that CA1 spikes are occurring nearly on their own, so how can they cluster?

 3. Finally, I have a few related questions about the PFC-CA1 interaction analyses. Namely, fig5b seems to show some PFC cells decrease their activity in relation to the occurrence of an isolated CA1 spike while others increase, is this the case? Could the authors quantify the number of PFC cells which show down-regulation vs up-regulation? Moreover, fig5c shows PFC activity changes at different time points leading up to the occurrence of an isolated CA1 spike - does the activity change increase as an closer to the time of an isolated spike (i.e. between theta cycles -4- 1 vs cycles 0-1)? Similarly, does the PFC prediction strength significantly increase at time intervals closer to cycle 0?

Reviewer #2: This paper reports an analysis of the out-of-field sparse spiking activity of hippocampal place cells. The authors examined such firing during locomotion as rats were running a maze in which they had to commute back and forth between several goal locations. They showed that isolated spikes recorded from place cells were coherently organized and were phase coupled to theta oscillations in a way different from the phase coupling observed when the cells fire in the place field. Furthermore the authors demonstrate isolated spikes in hippocampal cells are driven by prefrontal cortical activity. They conclude that out-of-field isolated firing is not noise but a signature of distributed and coherent information processing in the brain. 

This is an excellent paper that reports an original and very important finding. I have no major concerns. The question is relevant, the analyses are well done and the results are clear. 

I have only a few minor suggestions.

1) Although the present study focused on isolated spikes during locomotion and theta state, other studies have reported the existence of such out-of-field sparse firing during quiet alertness associated with theta state while the rat was waiting at a goal location (Hok et al., J. Neuroci. 1987). Some of the observations made in the Hok et al's paper were similar to those made here and it would be useful to cite this older study just to emphasize that isolated spikes might not occur only during locomotion. 

2) In a similar vein, it is worth mentioning the paper by Hasz and Redish (Hippocampus 2020) reporting a firing relationship between the medial prefrontal cortex and the hippocampus. This work shows that medial prefrontal activity influences hippocampal activity in a way that is consistent with the present observations.

3) It would be helpful to include some additional information about the key spatial features of the place cell sample used here, just to convince the reader that the analysis was done on "good" place cells and not a noisy sample. 

4) Really minor: 

* Line 144, add "and" before "its"

* Line 151, "place cells" instead of "place cell"

* Line 241, "similar" instead of "similarity"

* Line 383, it is unclear what "outside of a place cell's place and movement direction field" means

* Line 392, delete "during"

Reviewer #3, Zaneta Navratilova: This paper shows that hippocampal CA1 spikes that occur outside of "place fields" reflect coherent activity within the hippocampus, and are coordinated with activity in medial prefrontal cortex, showing that they cannot be characterized as local "noise." This important insight is shown with a well-designed and clear series of analyses. 

I have just a few comments to improve the paper. 

* The main text of the paper is well argued and clear. However, the figure legends are sometimes incomplete, and not clear enough for a non-expert reader. 

* Some supplementary figures (in particular figures S2 and S4) are very important controls, necessary to the conclusions of the paper, and the authors should consider including them as main figures. 

* Figure 5 compares PFC activity before and during theta cycles that include isolated spikes to similar theta cycles with no isolated spikes. Why not also compare the activity after isolated spikes? It is clear based on figure 6 that there is also a difference in PFC activity following isolated spikes, and it is not evident why the authors exclude this period only in figure 5. 

* In contextualizing this paper with previous studies in the literature, the authors miss a few relevant findings: in addition to different trajectories or future locations, non-local place cell spiking can also reflect different environments (Jezek et al., Nature, 2011) or reference frames, which are sometimes represented in alternating theta cycles, or simultaneously in the late phases of theta (Kelemen and Fenton, Plos Biology, 2010). In addition, spiking during travel in the non-preferred direction has been shown to occur more commonly and at higher rates during early stages of learning a specific trajectory through an environment (Navratilova et al., Frontiers in Neural Circuits, 2012). These findings suggest that "non-local" spiking can occur in different amounts, not just as isolated spikes. A discussion of whether the authors believe these are different phenomena (e.g. existing on a gradient of a single isolated spike to multiple adjacent or alternating theta cycles of non-local spikes), or if isolated spiking is a separate phenomenon, would be welcome. 

Minor edits:

* Figure S4 legend is grammatically unclear; should read: "Animal location, movement correlates and hippocampal network spectral signatures around excluded, adjacent and isolated spiking activity."

* Typo on line 241: should read: "very similar"

---

## [Editor Report · Decision Letter 2]

11 Aug 2021

Dear Jai,

Thank you for submitting your revised Research Article entitled "Prefrontal cortical activity predicts the extra-place field spiking of hippocampal place cells" for publication in PLOS Biology. I have now discussed your revision with other staff editors and with the Academic Editor as well. I am pleased to tell you that we will probably accept this manuscript for publication, provided you update your supporting data folder to provide the data for figures 4E and S3AB.

In addition, we would like you to consider changing the title to make it more accessible to a broader audience. We thought about the following but would be happy to discuss an alternative if you feel our suggestion misrepresents your findings or is inaccurate:

"Spiking of hippocampal place cells outside the place field reflects non-local spatial representations during spatial navigation."

We expect to receive your revised manuscript within two weeks. 

*Published Peer Review History*

*Early Version*

Sincerely,

Gabriel Gasque, Ph.D.,

Senior Editor,

ggasque@plos.org,

PLOS Biology

---

## [Editor Report · Decision Letter 3]

17 Aug 2021

Dear Jai,

On behalf of my colleagues and the Academic Editor, Jozsef Csicsvari, I am pleased to say that we can in principle offer to publish your Research Article "Prefrontal cortical activity predicts the occurrence of non-local hippocampal representations during spatial navigation" in PLOS Biology, provided you address any remaining formatting and reporting issues. These will be detailed in an email that will follow this letter and that you will usually receive within 2-3 business days, during which time no action is required from you. Please note that we will not be able to formally accept your manuscript and schedule it for publication until you have made the required changes.

**IMPORTANT: As you have noticed, I simplified the title, but kept the "prefrontal cortical activity" part, as I understand your argument. We think a simpler title is better for our broad audience. I would be happy to discuss more if you disagree. If you agree with our recommendation, please change the title as well in your uploaded document. 

PRESS

Sincerely, 

Gabriel Gasque, Ph.D. 

Senior Editor 

PLOS Biology

ggasque@plos.org